# Predicting neural responses to intra- and extra-cranial electric brain stimulation by means of the reciprocity theorem

Torbjørn V. Ness[1]*, Christof Koch[2], Gaute T. Einevoll[1,3]

**1** Department of Physics, Norwegian University of Life Sciences, Ås, Norway, **2** Allen Institute, Seattle, Washington, United States of America, **3** Department of Physics, University of Oslo, Oslo, Norway

* torbjorn.ness@nmbu.no

## Abstract

Electrical stimulation of nervous tissue is grounded in well-established biophysical principles, yet understanding its precise effect on neural activity remains challenging. This represents a major obstacle to both scientific and clinical applications of electrical brain stimulation.

We here show that the reciprocity theorem of electromagnetism can be applied more broadly than previously acknowledged, providing a framework for elucidating the effects of electrical brain stimulation on neurons. Through this new perspective, we account for the observed weak frequency-dependence of extracellular electrical stimulation–induced membrane potential changes, and demonstrate a $1/r$ falloff for nearby and $1/r^2$ falloff for distant neurons.

We also show that for transcranial electrical stimulation, the susceptibility of a neuron to the stimulation is directly proportional to the size of the current dipole moment resulting from somatic current input. The induced somatic membrane potential changes are small and exhibit weak spatial but strong orientation dependency across human neocortex.

The reciprocity-based approach to electrical stimulation reproduces a large body of previous experimental data, and provides new insights into how ES affects neurons.

## Author summary

Electric brain stimulation is widely used in neuroscience and medicine, from mapping brain function during surgery to treating disorders such as Parkinson's disease and depression. Yet predicting how stimulation alters neural activity has remained difficult, slowing progress in optimizing therapies and minimizing unwanted side effects.

**Data availability statement:** All simulation code is freely available through https://github.com/torbjone/ES_from_reciprocity.

**Funding:** T. V. N. and G. T. E. received funding from the European Union Horizon 2020 Research and Innovation Programme under Grant Agreement No., 101147319 [EBRAINS 2.0]. C.K. thanks the Allen Institute founder, Paul G. Allen, for his vision, encouragement, and support. The funders had no role in study design, data collection and analysis, decision to publish, or preparation of the manuscript.

**Competing interests:** C.K. holds an executive position and has a financial interest in Intrinsic Powers, Inc., a company whose purpose is to develop a device to assess the presence of consciousness.

We here show that a classical physics principle—the 170 years old Reciprocity Theorem—can be applied in a new way: Reciprocity allows us to directly predict, using a simple and efficient method, how neural membrane potentials respond to both intracranial and transcranial electric stimulation.

This method provides new insights into how electric stimulation affects neurons, reproduces a wide range of previous experimental results, explains recent counter-intuitive experimental findings and makes new testable predictions.

The reciprocity-based approach to electric stimulation enables directly optimizing for a cell-type and frequency-specific membrane potential response, which paves the way for more controllable electric stimulation.

## 1 Introduction

The reciprocity theorem (RT) is a foundational principle in physics, stating that in linear systems the source and the measurement sites can be swapped without a change in the measured signal. It was originally formulated more than 170 years ago in the context of electromagnetism [1], but has extended to a wide range of disciplines [2], including antenna design [3], biomedical imaging [4], earthquake seismology [5] and distance measurements in cosmology [6]. In the context of neuroscience, the RT has been used to swap the location of extracellular current sources—sometimes in the form of equivalent current dipoles—and extracellularly measured electric potentials [7,8]. Here we draw attention to the observation that the RT is also valid for intracellular current sources in the approximately linear (sub-threshold) domain. This opens up the possibility to directly studying how the somatic membrane potential is affected by extracellular current stimulation in a straightforward manner.

Extracellular potentials in neural tissue can be generated by exogenous currents, delivered either directly into extracellular tissue via invasive electrodes, referred to here as extracellular electric stimulation (ES) [9,10], or in a non-invasive manner, via transcranial electric stimulation outside the head (tES), which includes tACS, tDCS, and temporal interference (TI) [11]. Medicine and neuroscience have more than two centuries of experience with ES and tES [9,12], mapping the function of brain areas [12], studying functional connectivity [13,14], guiding awake neurosurgery [15] and treating neurological disorders such as Parkinson's disease, dystonia, depression, and epilepsy [9,16,17].

We have a reasonably good understanding of the biophysics underlying single neurons and how these generate extracellular potentials $V_e$, stemming from membrane currents propagating through an essentially ohmic extracellular medium [18–20]. Computer modeling of these mechanisms, based on the low-frequency, electro-quasistatic, ohmic-current-dominated approximation to the Maxwell equations, to study neural function is a mature field that has received substantial attention over the past decades [18–25].

The opposite situation, that is, how extracellular potentials influence sub-threshold and spiking activity is, on the other hand, insufficiently understood. This is true for

intracranial microstimulation [14,26–28], but even more so for ES that targets larger brain regions, like Deep Brain Stimulation (DBS) [9,17,29]. For tES, and, in particular, the emerging technique of TI [30–34], the causal mechanisms underlying any behavioral effects, above and beyond indirect effects (e.g., peripheral stimulation), remain controversial [11,35].

The basic biophysical mechanism underlying ES in the low kHz range has been well described in the literature [36–41]: A spatial gradient in the electric potential outside a neuron's morphology will affect the neuron's membrane potential. Given the known biophysics it is, in principle, straightforward to estimate the effect of ES via computational modeling [27,31,42–46]. However, given the daunting complexity of nervous tissue, together with the multi-dimensional nature of ES (geometry of stimulating electrodes relative to the distance and morphology of neurons, temporal-frequency content, amplitude, etc.), translating individual simulations into an intuitive understanding of how ES affects neurons has proven challenging.

This difficulty is well illustrated by the recent results of Lee et al. (2024) [28], where even single-cell subthreshold—and therefore presumably approximately linear—effects of ES in cortical slices are not easy to understand: They reported robust frequency-independent ES responses that were similar across cell-types, brain regions, and even species. This was surprising, because neurons are known to exhibit frequency-dependent behavior and have substantial cell-type specific variability in their morphological and electrophysiological properties. Neurons, therefore, have strong frequency- and cell-type dependent responses to intracellular current input.

This is where the *Reciprocity Theorem* (RT) can be helpful. It applies to Maxwell's equations of electromagnetism for time-invariant, linear media, in which the current is linearly related to the electric field. Intuitively, reciprocity implies that the relationship between an oscillating current and the resulting electric field is unchanged if the points where the current is placed and where the field is measured are interchanged [1,7,8]. The RT has often been applied when calculating extracellular potentials from current sources [47–50], or EEG signals from current dipoles [20,50–53]. The RT has also been used in the context of optimizing electric brain stimulation, in particular for tES [52–56]. Importantly, however, these previous usages of the RT did not explicitly consider neurons or their properties, only the neural tissue as a whole (Fig 1A and 1B).

The RT is, however, also applicable if the current source is intracellular. In our context, the RT states that for any tissue characterized by an electrical network consisting of ideal capacitances, inductances, resistances, and batteries, a change in the membrane potential $V_m(\mathbf{r}')$ due to an extracellular current source $I_{stim}$ at location $\mathbf{r}$ is equivalent to the change in the extracellular potential $V_e(\mathbf{r})$ due to a current input $I_{stim}$ at cellular location $\mathbf{r}$, (Fig 1C and 1D). This is convenient because it allows us to apply our understanding of how extracellular potentials are generated by neural activity [20] to better understand the effect of ES on the membrane potential of single neurons.

The RT is useful in systems with locations of special interest where one would like to know the measured signal resulting from sources at arbitrary locations. The cell body of a neuron is such a singular place. In the traditional approach to simulating ES, we simultaneously calculate the $V_m$ response at all cellular locations in response to an extracellular current source. However, experimentalists are often primarily interested in the somatic response to ES, both because this is by far the most accessible region of a neuron, and because the soma is electrically tightly coupled to the axon hillock where $V_m$ is transduced into action potentials that are the relevant output of any spiking neuron. That is, knowing the somatic $V_m$ is highly informative about the likelihood of that neuron generating one or more action potentials.

Let us compare the two approaches.

The standard framework has two steps; the first one involves calculating the extracellular potential resulting from the ES. The second step is simulating the effect on $V_m$ from the *spatial gradient* of this extracellular potential over the neural morphology. Both these steps can be numerically evaluated, but building a good, intuitive understanding is hard. In particular, the second step is opaque, and non-intuitive, even to long-time practitioners of the art, so that insights do not transfer well to other scenarios or parameters.

With intracellular somatic current input, the situation is easier—calculating the resulting membrane currents, and the resultant $V_e$ everywhere has been the goal of cable theory for the past century thanks to the efforts of Hodgkin, Huxley,

Traditional usage of the reciprocity theorem:
inferring extracellular potentials from extracellular stimulation

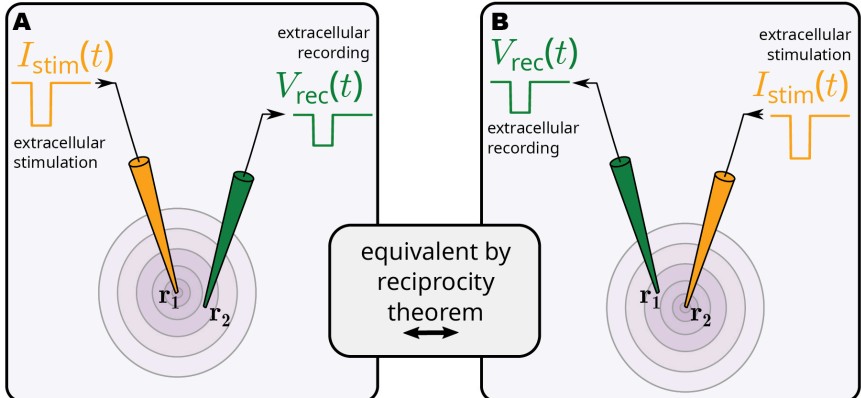

Novel usage of the reciprocity theorem:
inferring membrane potentials from extracellular stimulation

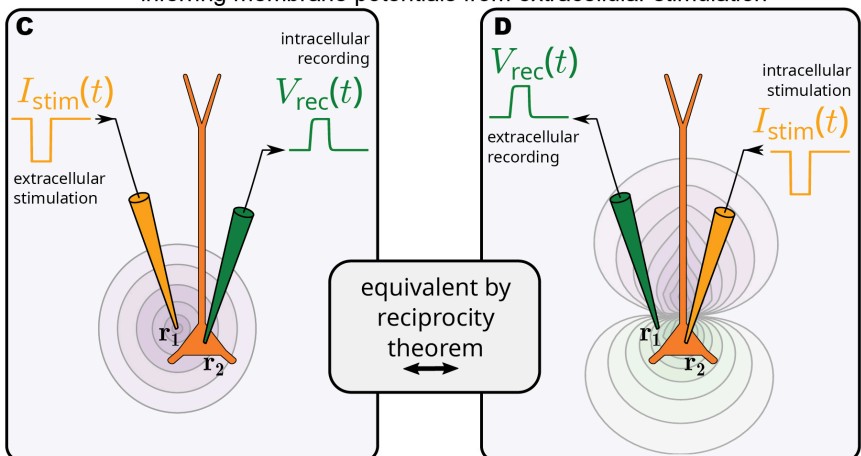

**Fig 1. The reciprocity theorem in the context of electric brain stimulation. A, B:** The traditional usage of the RT is to link extracellular current sources to extracellular potentials (or sometimes current dipoles to electric fields), without explicitly considering neurons and their properties [52–56]. **C, D:** We here point out that by moving the current source into the cell the RT can be used to directly account for the effect of ES on membrane potentials. Note that the intracellular stimulation current in panel D should be treated as part of the membrane current (see Methods).

Cole, Rall, Jack, Noble, Tsien and others, with a well-developed calculus and intuition [20,57,58]. From a somatic intracellular current source, we simultaneously compute $V_e$ at all relevant extracellular locations. Through the RT, we therefore know the somatic $V_m$ response to ES from any of these locations. The RT-based approach therefore allows us to take a *soma-centric viewpoint* that is advantageous.

The RT assumes linearity, and since neurons often behave approximately linearly in the subthreshold regime [59–61] (as we will demonstrate later), the RT can be applied to study the subthreshold effects of ES. Thus, if we can calculate the extracellular potential resulting from somatic current input, we know the somatic membrane potential response to the same extracellular current (Fig 1). Perhaps counterintuitively, the RT is also, as demonstrated here, germane to tES as practiced today, where ES current flow is limited to 4 mA or less [62].

We here first show how the RT is applicable to accurate simulation of ES on biophysically-detailed and simplified models of cortical and subcortical neurons. We demonstrate how this approach aids in understanding of ES on different neural

elements, and how this can explain many empirical findings, and make new predictions. We then apply the RT to infer the effect of tES on single neurons, deep inside the human brain.

## 2 Results

### 2.1 Validating the reciprocity-based approach

We consider the effect of an extracellular sinusoidal current $I_{stim}$ at location $\mathbf{r}_{EC}$ on the somatic $V_m$ of a canonical rat cortical layer 5 pyramidal cell with detailed dendritic morphology and ten voltage- and calcium-dependent membrane conductances distributed through the cell (Fig 2A) [63].

In accordance with the RT, the model shows that $V_m$ induced by ES—here a current of 1 µA and 10 Hz induced a peak $V_m$ of 2.7 mV (Fig 2B)—is indistinguishable from $V_e$ at location $\mathbf{r}_{EC}$ due to intracellular current input $I_{stim}$ to the soma (Fig 2B and 2C, "reciprocity-based").

Amplitudes for ES are, however, typically large compared to intracellular currents (mA vs. nA) which would trigger hyper-excitability if delivered intracellularly to active cell models. To readily apply the RT to active cell models, we avoided this methodological issue in Fig 2 by using a fixed subthreshold current amplitude for the intracellular current to the active cell model, and re-scaling $V_e$ to reflect the current amplitude used for the ES. That is, while ES is applied directly to the fully active cell model, the reciprocity-based approach is a linear extrapolation of the subthreshold response of the active

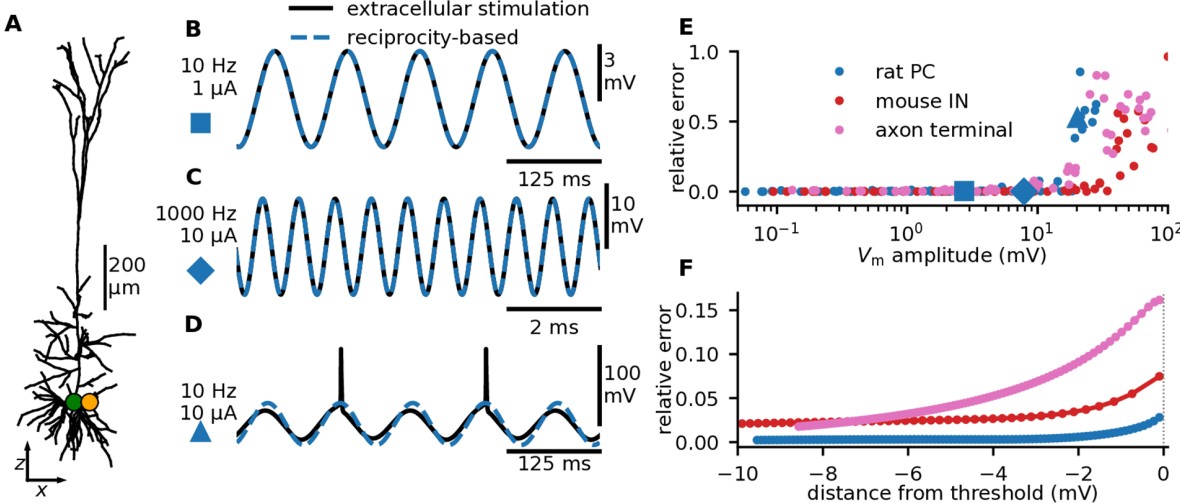

**Fig 2**. **Applying the reciprocity theorem to subthreshold ES of a compartmental model of a cortical neuron. A:** Stimulating a canonical cortical layer 5 pyramidal cell (PC) model [63] with a nearby current (orange dot, 50 µm outside the soma), while recording the somatic membrane potential (green dot). **B:** $V_m$ responding to ES of 10 Hz and 1 µA (black line). The reciprocal situation, *i.e.*, injecting the same current into the soma while measuring changes in $V_e$ at the orange dot, leads to indistinguishable results (blue dashed line). Note that the reciprocity-based solution predicts the membrane potential response, that is, the deviation from the baseline, but not the resting membrane potential itself. **C:** Same as in B, but for a 1000 Hz, 10 µA current. **D:** Same as in B, but for a 10 Hz, 10 µA current. This combination of amplitude and stimulation frequency evokes action potentials, and is therefore outside the linear regime. **E:** The relative error as a function of $V_m$ at the stimulation frequency, for three different models: the layer 5 pyramidal cell from panel A, a mouse cortical layer 5 GABAergic PV interneuron (IN), and a straight, unbranched myelinated axon with periodic nodes of Ranvier (see Methods). We injected different currents (0.1, 0.5, 1, 5, 10, 50, 100 µA) at different frequencies (1, 10, 100, 1000 Hz) and distances (25, 50, 100 µm) from the target compartment (soma for the PC and IN model; closest axon terminal for the axon model). The instances of panels B, C, and D, are marked, respectively, by a large blue square, diamond, and triangle. The relative error is defined as $\mathrm{SD}(V_{ES} - V_{reciprocity\text{-}based})/\mathrm{SD}(V_{ES})$. **F:** For each cell model 100 simulations were executed with small increments in the ES amplitude, where the range of amplitudes was chosen so that all models spiked for the highest amplitude. The highest subthreshold amplitude response in the target compartment was identified, and the spike threshold was defined as 0.1 mV above this value. The relative error of applying the RT-based approach is plotted versus the distance of the resulting membrane potential response from spiking threshold (only including subthreshold simulations). The ES was 50 µm outside the target compartment and had a frequency of 100 Hz.

cell model (see Methods). This approach can be expected to be accurate when the membrane potential response from ES is within the linear regime. Indeed, the active cell model effectively responds linearly to ES, in line with experimental observations [59,61]. This remains true if ES is increased to 10 µA with a frequency of 1 kHz (Fig 2C), giving a peak $V_m$ of 7.8 mV from rest (Fig 2C).

However, the closer $V_m$ veers to the cell threshold, the less applicable the RT will be. Thus, for ES that directly evokes spikes (here a current of 10 µA at 10 Hz), the RT approach is not reliable (Fig 2D). We quantify the relative error of applying the reciprocity-based approach. This error is zero for passive cell models, a few percent for active models if the evoked $V_m$ < a few mV (Fig 2E, dots) but abruptly increase for ES amplitudes that are driving the values of $V_m$ close to or above the threshold for spiking. The relative error is 1.1%, 4.6%, and 12% for the pyramidal cell, the interneuron, and the axon terminal models, respectively, 1 mV below the spiking threshold, and reduces to 0.3%, 2.5%, and 4.0% at 5 mV below the spiking threshold (Fig 2F).

We thus confirmed that the RT-based approach gives similar numerical predictions to the traditional approach in the approximately linear subthreshold regime. We can also compare these numerical predictions to measurements from the experimental literature: Lee et al. (2024) [28] recorded somatic $V_m$ responses to ES of 100 nA, at a distance of 50 µm from the soma (in the frequency range 1–100 Hz) of about 0.3 mV. How does this compare to our results? At 10 Hz and a distance of 50 µm, a 1 µA current amplitude yields a $V_m$ of about 3 mV (Fig 2B). Extrapolating linearly, the predicted $V_m$ of an ES of 100 nA will be 0.3 mV, very similar to Lee et al. (2024) [28]. This serves as a qualitative check, confirming that our results are biologically plausible.

## 2.2 RT for studying the effects of ES on different neural elements

The effect of ES on the somatic $V_m$ is informative about how the neuron's spiking activity will be affected (while spike initiation happens at the axon initial segment, it is electrically closely coupled to the soma). Via the RT this can be studied by considering $V_e$ from a somatic current input. This is convenient because we have a good understanding of how neurons generate extracellular potentials [20,22]. We therefore review some general features of extracellular potentials generated by current input to neurons, and consider the implications for how neurons are affected by ES.

For clarity of exposition, we restrict ourselves to passive cell models, but as demonstrated in Fig 2, our results are directly relevant for active cell models operating in the subthreshold regime.

**2.2.1 Pyramidal cells.** We consider a passive version of the cortical rat layer V pyramidal cell (PC) model from Hay et al. (2011) [63] and calculate $V_e$ resulting from somatic white-noise current input. All frequency components of the white-noise current have an equal amplitude of 1 nA (Fig 3A).

The amplitude of a frequency component of $V_e$ at any given location corresponds, through the RT, to the amplitude of $V_m$ caused by a sinusoidal ES at that particular frequency and location. Note that we have effectively quantified the subthreshold ES response of the neuron model for any ES location through a single simulation with $V_e$ calculation. For the traditional approach to simulating ES, each ES location requires a separate simulation.

The $V_e$ from somatic current input has a dipole-like shape, aligned with the orientation of the apical dendrite, perpendicular to the cortical surface (Fig 3A), as expected from theory [20]. The negative regions of $V_e$ correspond to where $V_m$ is 180 degrees "out of phase" with the ES, that is, where $V_m$ will be positive (depolarizing) in response to the negative phase of ES. This is again consistent with literature [28,64]. ES located in the crossover region between the negative and the positive regions of $V_e$ will have a negligible effect on the somatic $V_m$, while for ES in the positive region outside the apical dendrite, the somatic $V_m$ will "be in phase" with the ES, that is, the $V_m$ response will be positive to the positive phase of ES.

ES will always depolarize one region of a neuron, while hyperpolarizing others [45,46,64–66]. Indeed, it is impossible to devise an ES that will have a purely depolarizing or hyperpolarizing effect across an entire neuron. This can also be shown through the RT: Current conservation ensures that at any given time, a neuron's membrane currents must sum to

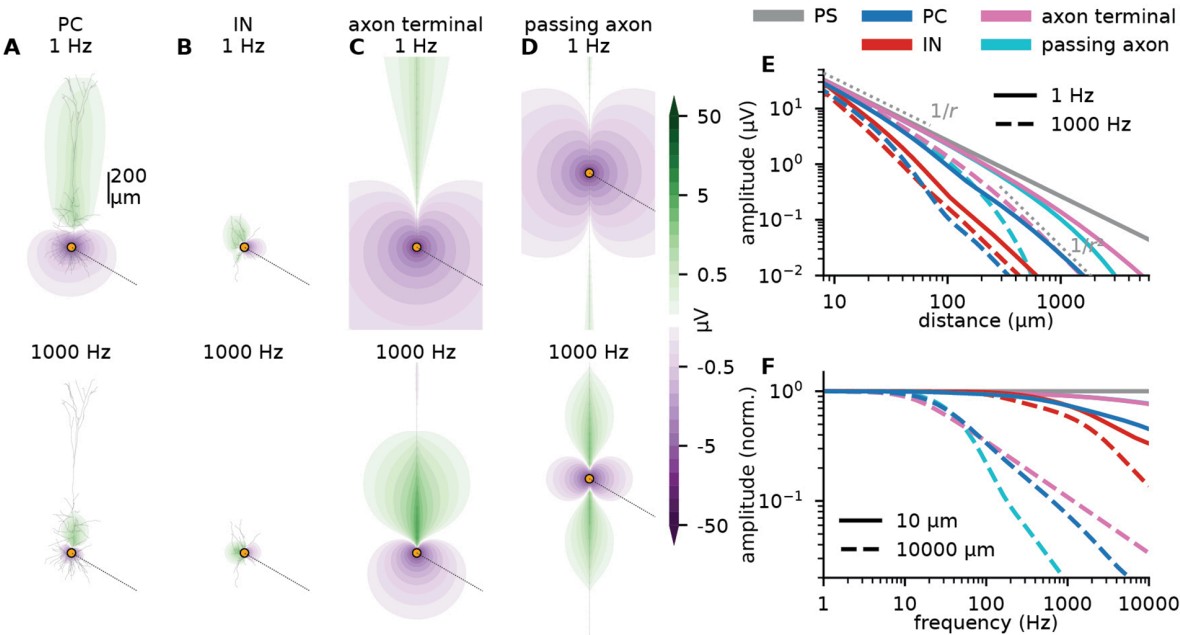

**Fig 3**. **Extracellular potentials surrounding different neural models are informative about the effect of ES. A:** $V_e$ following somatic white-noise current input, where each frequency component has an amplitude of 1 nA (see Methods), to a passive rat cortical layer 5 pyramidal cell (PC) model (passive version of the model shown in Fig 2A). $V_e$ is calculated on a dense grid around the neuron. Through Fourier analysis, the 1 Hz (top) and 1000 Hz (bottom) components of $V_e$ are extracted (see Methods). **B:** Same as in panel A, except for a passive mouse cortical layer 5 interneuron model (IN). **C:** Same as in panel B, except for a passive myelinated axon model, with the current input at the terminal end. The axon is oriented vertically, parallel to the apical dendrite in panel A. **D:** Same as in C, except that the current input is in the middle of the axon. **E:** $V_e$ as a function of distance from the site of current input, calculated along the dotted lines in A-D (at -30° relative to the horizontal direction). Different colors correspond to the four different scenarios in panels A-D, as well as $V_e$ from a point source (PS). Full and dashed lines correspond to 1 Hz and 1000 Hz input respectively. **F:** $V_e$ as a function of frequency for white noise current input, calculated at two different distances along the dotted lines in A-D. Full and dashed lines correspond to 10 µm and 10,000 µm distance respectively. All amplitudes are normalized to the value at 1 Hz.

zero, and extracellular potentials are therefore intrinsically dipole-like in nature, with both positive and negative regions. A current input to the soma or to the apical dendrite will give rise to dipole-like extracellular potentials of opposite orientation [20,67,68]. As a consequence, ES will have opposite effects on somatic and apical dendritic $V_m$.

Interestingly, in analogy to how pyramidal neurons are expected to dominate LFP/EEG/MEG signals because they are spatially aligned [20,23,69], this geometrical alignment also predicts that pyramidal cells with approximately the same orientation and distance from an extracellular current source will be affected in the same manner. We can therefore expect distant ES to have a synchronizing effect on pyramidal neurons next to each other with the same orientation (*i.e.*, in the same gyrus; [45]).

The $V_e$ from somatic current input becomes more spatially confined at higher frequencies (Fig 3A), as expected from intrinsic dendritic filtering [20,67]. This suggests that, in general, low-frequency ES will have a stronger effect on $V_m$ than high-frequency ES. Importantly, however, $V_e$ close to the soma is relatively frequency-independent below about 100 Hz (Fig 3F). This surprising result—the effect of ES on near-by cell bodies is approximately frequency independent—was reported—but left unexplained—in experimental ES work by Lee et al. (2024) [28].

As earlier noted, in the traditional approach to simulating ES, each ES location requires a separate simulation, which has made general insights regarding the distance-dependence of ES hard to extract. Through the RT we can, however, draw from the well-established theory of extracellular potentials, and how they, for example, decay with distance from the

electrode. To understand how ES decays with distance, assuming an infinite, homogeneous and isotropic volume conductor, we approximate $V_e$ at location $\mathbf{r}$ for a model with $N$ compartments via the point-source approximation [20,22],

$$V_e(\mathbf{r}, t) = \sum_n^N \frac{I_{m,n}(t)}{4\pi\sigma|\mathbf{r} - \mathbf{r}_n|} ,$$ (1)

where $I_{m,n}$ is the membrane currents of compartment $n$, $\sigma$ is the extracellular conductivity, and $\mathbf{r}_n$ is the position of compartment $n$.

Sufficiently close to a soma (at $\mathbf{r}_s=0$), $V_e$ will be dominated by the somatic membrane current alone [20,70],

$$V_e^{near}(\mathbf{r}, t) \approx \frac{I_{m,s}(t)}{4\pi\sigma|\mathbf{r}|} .$$ (2)

This "monopole"-like approximation is accurate as long as $|\mathbf{r}| \ll$ the frequency-dependent length constant [20], and the region where $V_e \propto 1/|\mathbf{r}|$ is valid can be roughly estimated from Fig 3E. In absence of other inputs, the somatic membrane current itself will be dominated by the input current which counts as part of the membrane current $I_{m,s} \approx I_{stim}$,

$$V_e^{near}(\mathbf{r}, t) \approx \frac{I_{stim}(t)}{4\pi\sigma|\mathbf{r}|} ,$$ (3)

confirmed in Fig 3E (blue versus gray line). It follows that in the near-field limit, the amplitude of $V_e$ will be approximately frequency-independent, and that for near-by ES, the specific membrane properties are of limited importance for determining the response of $V_m$. The dominating factor is the ES current itself, as seen in our simulations for different cell models (Fig 3E).

Sufficiently far away from the neuron, we can express $V_e$ in terms of the current-dipole approximation [20],

$$V_e^{far}(\mathbf{r}, t) \approx \frac{p_s(t)\cos\theta}{4\pi\sigma|\mathbf{r}|^2},$$ (4)

where $p_s$ is the current-dipole moment of the cell resulting from the somatic current injection, and $\theta$ is the angle between the dipole orientation and the measurement point. In this far-field regime, $V_e$ decays with distance as $1/r^2$ ("dipole"-like) [20,70], and has a clear, albeit still relatively weak, frequency dependence above about 10 Hz (Fig 3F). From use of the RT, these properties are expected to apply to responses of $V_m$ to distant ES as well.

Interestingly, we deduce that for distant ES—like tES—the somatic $V_m$ response of a neuron is proportional to the current dipole moment of that neuron in response to somatic current input $p_s$. Cellular features which maximize $p_s$ will therefore also maximize the ES sensitivity. To give an example of how this insight might be useful: Vieira et al. (2025) [71] demonstrated that a small subgroup of cells were particularly strongly affected by tACS. Our results predict that this subgroup of neurons should exhibit particularly strong current-dipole moments from somatic current input, which is straightforward to simulate. This can potentially be used to identify this particularly tACS-sensitive population by scanning through data bases of cell models.

**2.2.2 Interneurons** Somatic current to a mouse layer V PV-interneuron model results in a $V_e$ with a dipole-like shape (Fig 3B), whose orientation depends on the geometry of dendrites. Since interneurons lack apical dendrites, the effect of ES on a population of interneurons will be more variable than for pyramidal neurons, and we cannot *a priori* expect the ES to have a synchronizing effect [45,67,68].

Lee et al. (2024) [28] reported $V_m$ of similar amplitudes in pyramidal neurons and interneurons responding to nearby ES. This is consistent with our observations (Fig 3E). However, at least for low frequencies, $V_e$ from interneurons enter

the far-field regime at distances closer to the soma than for pyramidal neurons [70]. Thus, in this regime, ES on pyramidal cells will be stronger than on inhibitory neurons for ES that is more than some tens of micrometers away from the somas (Fig 3E).

$V_e$ for the interneuron is almost independent of frequency below 1000 Hz (Fig 3F), as it is too compact for intrinsic dendritic filtering to have any low-pass filtering effect [67]. These results indicate that for distant (e.g. transcranial) ES, low-frequency stimulation will have a predominately excitatory effect (Fig 3E, solid lines, blue versus red), while high-frequency ES will be more balanced, or even biased towards having an inhibitory effect (Fig 3E, dashed lines, blue versus red), in line with experiments [34].

**2.2.3 Axons.** Axon terminals are known to be the most sensitive neural element to ES [26,27,72], and furthermore, it has recently been demonstrated that subthreshold polarization of axon terminals can strongly affect neurotransmitter release [46]. It is therefore often of interest to know the $V_m$ response of axons to ES or tES. We therefore consider a current input to an axon end or terminal (Fig 2E). While this is difficult to achieve in practice due to their minute size, this is irrelevant for the validity of the RT.

$V_e$ resulting from the current input has a dipole-like shape (Fig 3C), falling off as $1/r$ close by, and as $1/r^2$ far away (Fig 2E). The amplitude of $V_e$ is higher than for the same input to cell bodies (Fig 3E). According to the RT, the same should hold for the amplitude of the $V_m$ response to ES, compatible with both experiment [72] and modeling [27].

Interestingly, $V_e$ following current input to the middle of the axon model (Fig 3D) gives rise to "quadrupole-like" shape, with the effect of ES decaying as $1/r^3$ in the far-field [20]. That is, axon terminals are more excitable to ES than passing axons, although the difference is only visible beyond a certain distance (Fig 3E).

## 2.3 Analytical investigation of ES in simple cell models

To understand complex physical systems, it helps to start with tractable models capturing the essential underlying principles. Such simple models for the effect of ES on neural activity have been lacking. Through the RT we can, however, draw from analytic solutions of $V_e$ from somatic current input to a passive ball-and-stick model, and directly apply the resulting formulas to investigate how the neuron model is affected by subthreshold ES. Near the soma, $V_e$ is dominated by the somatic membrane current alone (Eq (2)), while sufficiently far away the current-dipole approximation holds (Eq (4)). To derive an analytical expression for how a neuron is affected by ES we need expressions linking an arbitrary somatic input current to the resulting somatic membrane current for the near-field case, and to the resulting current-dipole moment for the far-field case.

Pettersen et al. (2014) [73] derived expressions for somatic membrane currents $\mathbf{T}_I^s$, and the dipole moment $\mathbf{T}_P^s$. For an arbitrary input current $\mathbf{I}_{in}$ (for example, white noise), $\mathbf{I}_s = \mathbf{T}_I^s \mathbf{I}_{in}$ and $\mathbf{P}_s = \mathbf{T}_P^s \mathbf{I}_{in}$ (boldface symbols signify complex numbers). Using these expressions (see Methods) confirms that $V_e$ decays as $1/r$ in the near-field and as $1/r^2$ in the far-field regime (Fig 4A).

In the near-field, the ball and stick neuron is relatively insensitive to the frequency of the ES up to about 100 Hz, decaying as $1/\sqrt{f}$ at high frequencies (Fig 4B-4G, top row). Far away, the response is frequency-independent for low frequencies, eventually decaying as $1/f$ (Fig 4B-4G, bottom row).

ES is relatively insensitive to changes in the neuron's electrical properties (Fig 4B-4D): in particular, the value of $R_m$ is largely irrelevant. This may seem surprising: The somatic $V_m$ response to current input is highly $R_m$-dependent as can be understood from Ohm's law, $V = RI$. $V_e$, however, is fully determined by the membrane current; for a fixed current input, $R_m$ therefore only plays a secondary role in the spatial distribution of the return currents.

For near-by ES, the length of the dendrite $l$ is unimportant, because $V_e$ is dominated by the somatic current alone. The effect of the dendritic length is more pronounced for distant ES (Fig 4E), since the length of the stick will directly affect the current-dipole moment through affecting the distance between the sinks and sources [70]. Interestingly, these results indicate that the spatial extent of neurons is primarily important for distant, low-frequency ES. This is also easily understood

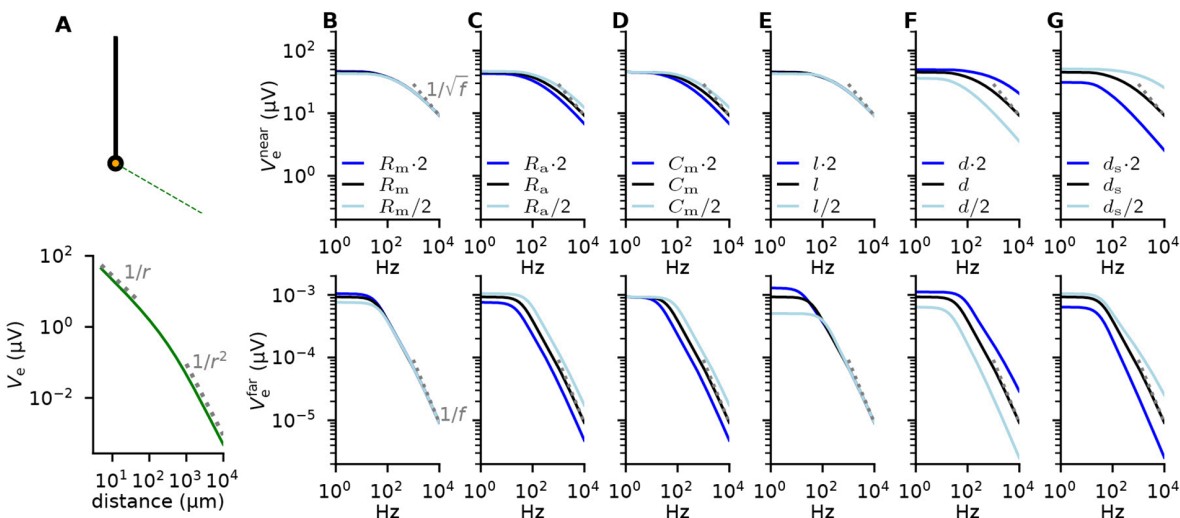

**Fig 4. Analytical expressions for $V_e$ gives insight into the parameter-dependence of ES. A:** Illustration of how $V_e$ decays with distance from the soma of a ball and stick neuron (see schematic) receiving a somatic current input with a frequency of 1 Hz and an amplitude of 1 nA. The bottom panel shows results from a numerical simulation of changes in $V_e$ with increasing distance. Grey dotted lines indicate expected trends in the near ($1/r$) and far-field ($1/r^2$) regimes. **B-G:** The amplitude of $V_e$ as a function of frequency for a white-noise input current where each frequency component has an amplitude of 1 nA, and $V_e$ is measured either very near (top) or very far away from (bottom) the soma. The three lines correspond to different parameter choices for a single parameter (columns), either default (black), increased by a factor of two (blue), or decreased by a factor of two (light blue). The parameters are membrane resistance $R_m$ in B, axial resistance $R_a$ in C, membrane capacitance $C_m$ in D, length of the stick $l$ in E, dendritic diameter $d$ in F, and somatic diameter $d_s$ in G. The grey dotted lines show the expected trends in the high-frequency limit in the near- ($1/\sqrt{f}$) and far-field ($1/f$) [73].

by considering the current-dipole moment for somatic current input: For high frequencies, the frequency-dependent length constant [58], which determines the spatial distribution of return currents, becomes much smaller than the total cell length, rendering the total length irrelevant at these frequencies.

The effect of ES is most sensitive to the diameter of the dendrite $d$ (Fig 4F-4G). For an increased $d$, the effect of ES increases (Fig 4F), as expected from the literature where it has been reported that large-diameter axons are most sensitive to ES [27,36,65,74]. The opposite is true for soma size $d_s$, where the effect of ES decreases with an increased diameter (Fig 4G).

The opposite effect of the somatic and dendritic diameter is hard to make sense of through the traditional way of thinking about ES. However, through the RT-based approach it makes sense: We know that for somatic current input, a bigger soma will lead to more local and less distributed return currents, thereby resulting in a weaker current dipole moment and a lower amplitude $V_e$. Through the RT we therefore know that a larger soma will decrease the sensitivity of the somatic $V_m$ to ES. A thicker apical dendrite will have the opposite effect of less local and more distributed return currents, thereby resulting in a stronger current dipole moment.

This illustrates that in the traditional approach to simulating ES it is hard to build an intuitive understanding of how any neural property will affect how a neuron is affected by ES, but that the RT-based approach offers a new perspective on ES which can be helpful in this regard.

## 2.4 Intrinsically active cells

We demonstrated that the RT-based approach works well in the approximately linear subthreshold regime (Fig 2), however, neurons are—due to their ability to fire action potentials—fundamentally non-linear entities. Some classes of

neurons also exhibit intrinsic spiking activity, like in the subthalamic nucleus (STN) of the basal ganglia which is an important target for DBS therapy in patients with Parkinson's disease [75].

Note that even though the effect of ES on neural dynamics can be non-linear, the direct, isolated effect of ES on membrane potentials usually is linear: In the traditional way of simulating, the ES enters as an additive term in the cable equation, $(1/r_i)\partial E_{||}(x,t)/\partial x$, which only depends on the electric field along the neurite $E_{||}$ and the cable properties, and not on the neural membrane potential. The ES-induced effect itself on the membrane currents is therefore linear and additive, even when the resulting membrane potential and network dynamics is not. The RT-based approach might therefore be able to isolate the linear effect of the ES itself on the membrane potential, which can then be added to a non-linear model.

To investigate this, we considered a macaque STN neuron (Fig 5A), which intrinsically spikes at around 30 Hz (Fig 5B). We introduced an ES with parameters arbitrarily chosen to have an amplitude of 100 µA at 10 Hz, located 500 µm away from the soma (Fig 5C). The spiking behavior of the cell was substantially modulated by the ES (Fig 5D versus Fig 5B). Since the behaviour of the membrane potential is clearly non-linear, the RT can not be used directly; however, the effect of an ES can still be estimated, by inserting the ES current (Fig 5C) into the soma of a passive version of the cell model, and calculating the resulting $V_e$ at the stimuli location: Through the RT, this corresponds to the isolated ES-induced response of the somatic membrane potential $\Delta V_m$ (Fig 5E). We then introduce this effect into the active intrinsically spiking cell model (instead of the ES) as the somatic current input that in isolation would cause this $\Delta V_m$. This gave rise to a very similar ES-modulated activity as the traditional approach (Fig 5F versus Fig 5D).

The example above illustrates that the effect of the ES can——at least in this case——be isolated and then added to a non-linear model. This is relevant for point neuron simulations of deep brain stimulation of the basal ganglia: Through the RT and using biophysically detailed cell models, we can efficiently and easily estimate how different cell-types at different locations are affected by ES. The non-linear effect of the ES at the network-level (with intrinsically spiking neurons) can then be investigated by highly efficient point-neuron, or potentially even mean-field network models.

## 2.5 More complex extracellular mediums

Thus, the RT-based approach is helpful for analyzing the effect of ES on different neural elements embedded in an infinite, homogeneous, and isotropic medium. Of course, real brains are none of those [76–78]. However, as long as nervous tissue behaves approximately linearly, that is, varying the applied current varies the induced electrical field proportionally, the RT remains relevant, including when the extracellular conductivity is inhomogeneous (position-dependent), anisotropic (direction-dependent), or frequency-dependent. In this section, we demonstrate how we can use the RT for more complex extracellular mediums, and how to leverage the RT to gain additional insights into ES from previous research that explored extracellular potentials.

For extracellular mediums with anisotropies or simple planar inhomogeneities in the extracellular conductivity, only minor changes are needed to the formalism for calculating $V_e$ [20,49]. Taking such effects into account is easy with appropriate simulators like `LFPy` [79].

Cortical tissue is moderately anisotropic and inhomogeneous [80]; however, it has been shown that the effect on intracortical potentials is negligible [49,77,81], implying that we are justified in disregarding such effects for intracortical ES as well.

For measurements of potentials at the cortical surface (ECoG), the membranes and external materials covering the cortex can have a substantial effect on the recorded potentials [49,82,83]: Electrically insulating materials (like non-conducting mineral oils or air) amplify signal amplitudes, while highly-conducting materials (like saline or metal) reduce signal amplitudes [20,82,83]. ES at the cortical surface will therefore have a larger effect on neural activity if a non-conducting material is used as a cover material.

It can be important to take the presence of the recording/stimulating equipment on extracellular potentials into account, in particular if it accounts for a large non-conducting volume that affects $V_e$ [47,49,84–89]. This necessitates numerical

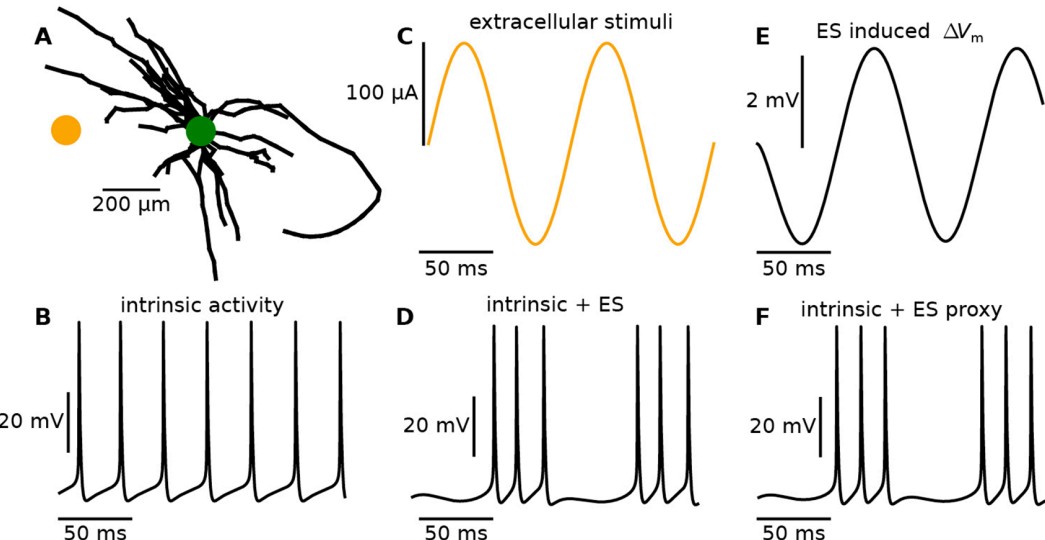

**Fig 5. Intrinsically active cell model. A:** Macaque STN neuron from Miocinovic et al. (2006) [43]. **B:** The intrinsic spiking activity of the model in the absence of input. **C:** An electric stimulation (location marked by orange dot in panel A) at 10 Hz with a current amplitude of 100 µA, 500 µm away from the soma of the cell model. **D:** The somatic membrane potential of the cell under the influence of the ES in panel C, simulated through the traditional approach. **E:** Through the RT, we can calculate the effect of the ES in isolation, by inserting the current in panel C into the soma of a passive version of the cell model, and calculating $V_e$ at the ES location. The resulting amplitude of $V_e$ was 2.1 mV. From the same simulation, we can calculate the input impedance of the model, and from this the equivalent somatic current amplitude of 67.6 pA needed to cause this 2.1 mV oscillation. **F:** The original active cell model is stimulated with an "ES proxy", that is, by injecting this current into the soma (amplitude 67.6 pA) designed to mimic the $\Delta V_m$ in panel E. This closely approximates the direct simulation (panel D).

methods like the finite element method; yet, this does not invalidate the RT. Thus, Buccino et al. (2018) [86] demonstrated that while microwires hardly influenced $V_e$, the presence of the much larger Neuropixels and Neuronexus MEA probes boosted $V_e$ from neurons in front of them by almost a factor of two, and decreased the signal amplitude from neurons behind them by about a factor of two (a shadowing effect). If the recording electrodes are instead used to inject currents, we can expect similar conclusions for the resulting membrane potential of nearby neurons.

The examples in the previous paragraphs highlight how the RT-based approach supplies a fresh perspective on ES, allowing us to take advantage of earlier research and insights.

### 2.6 Transcranial electric stimulation (tES)

Electroencephalography (EEG) signals, recorded on the scalp, are strongly influenced by the variable conductivities of cortical grey matter, cerebrospinal fluid, skull, scalp, and musculature [25,50,76,78,90]. EEG signal prediction therefore warrants complex volume-conductor models, called *head models*. Importantly, all such models are linear [18,19]. This enables us to apply the RT to infer changes in $V_m$ responding to tES, that is, current stimulation from the scalp: we use a head model to calculate the EEG signal at a given location on the head ($V_e$) due to a somatic current input to a neuron (operating in an approximately linear regime) at a given location in the brain. Through the RT, this corresponds to $V_m$ of that neuron from tES at the EEG electrode location.

For calculating EEG signals from neural activity, it is common to represent neural activity as equivalent current dipoles. Calculating these from simulated neural activity is straightforward for multi-compartment models [79]. Furthermore, the link between current dipoles and the resulting EEG signals is well developed [18–20], and current dipoles can be used with simple or detailed head models [68]. The EEG signal generated by a single current dipole $\mathbf{P}_n(t)$ can, in general, be expressed as $V_e(\mathbf{r}, t) = \mathbf{M}(\mathbf{r}, \mathbf{r}_n)\mathbf{P}_n(t)$, where $\mathbf{M}$ is the so-called lead-field matrix.

To illustrate the application of the RT to tES, we simulate a 10 Hz, 1 nA somatic current input to a passive L5 pyramidal cell model and calculate a resulting current-dipole moment amplitude of 186 nAµm (Fig 6A-6C). Note that we here only consider the *z*-component, set to be along the apical dendrite, perpendicular to the cortical surface. The rationale is that because of rotational symmetry for pyramidal cells around the *z*-axis, the other components of the current-dipole moment will cancel for a population of pyramidal cells.

We also quantify the relationship between the frequency of the input current and the resulting current-dipole moment amplitude for four different cell models (Fig 6D). For a ball and stick model, the current-dipole moment is frequency-independent for low frequencies, falling off as $1/f$ at higher frequencies (Fig 4), while the decay is somewhat less steep for the other three models. For the pyramidal cell, the amplitude decays from 198 nAµm at 1 Hz to 8.6 nAµm at 10 kHz, that is, by a factor of 23 for a frequency increase over 4 orders of magnitude. The axon terminal is about an order of magnitude more sensitive to tES than the pyramidal cell, consistent with previous simulation results [45,91].

To calculate the signal on the scalp evoked by this particular neuron, we use the New York head model, as it is a detailed model with a freely available lead field for each EEG electrode (see Methods). The amplitude of the evoked potential at a simulated EEG electrode above the middle frontal gyrus (Fig 6E) is 38 pV (Fig 6F), decaying with distance from the dipole, and changing sign, as expected from a dipole (Fig 6G). Of course, such an tiny evoked signal would not be detectable, given the ambient biological and instrumental noise levels, but constitutes an idealized estimate.

Via the RT, this implies that a 10 Hz, 1 nA current injection through the same EEG electrode will induce 38 pV change in $V_m$. Common current amplitudes for tES are on the order of 1 mA [11,53,92,93], a million times bigger. That is, a 10 Hz, 1 mA tES stimulus will induce a $V_m$ amplitude of 38 pV $\times 10^6$ = 38 µV (Fig 6H-6K). A million-fold scaling factor might at first glance seem questionable, but note that the only relevant constraint is that the resulting $V_m$ should be subthreshold, and 38 µV remains well within the linear regime (Fig 2E). Even a ten times larger current of 10 mA, above the safety level for human usage [62], would induce an evoked $V_m$ of only a third of a millivolt.

Our results indicate that tES in the range of 1-4 mA on the scalp induces membrane potential changes of 38-152 µV at the soma of a single pyramidal cell inside the human brain. This is consistent with previous findings: 1-4 mA tES gives a 0.21 - 0.84 mV/mm field in the human brain for the head model we use, and similar values are reported experimentally by others [94,95]. In rat hippocampus, somatic depolarizations on the order of 0.2 mV per V/m of applied field have been observed [11,59,96]. For the field values listed above (0.21 - 0.84 mV/mm), this corresponds to 42-168 µV somatic depolarization, in line with our results. Note, however, that even though detailed head models have been demonstrated to be quite accurate for calculating electric fields from tES [94], there is considerable variability in both in the anatomy of individual human brains, and in their conductivity values. The electric fields from tES will therefore also be person-specific [97]. The numbers presented here should therefore be viewed only as plausible examples.

The example in the previous paragraph treated a single cortical location and many EEG electrodes, but it is equally possible to focus on a single electrode and evaluate the $V_m$ response for the given cell model at any cortical site. For the somatic 10 Hz, 1 nA input to the pyramidal cell model, the amplitude of the resulting current-dipole moment was 186 nAµm. By multiplying this number with the lead field $\mathbf{M}(\mathbf{r}, \mathbf{r}_n)$, and further scaling it by $10^6$, we obtain the subthreshold somatic $V_m$ response at any cortical location (Fig 6J).

Cortical regions where the normal points towards the electrode typically experience the opposite effect relative to regions facing away from the electrode (Fig 6J). As expected, tES is strongest close to the electrode, but overall its effect is quite evenly distributed across the brain. Although the amplitude will be frequency dependent (Fig 6D), the spatial distribution will not be, since this term comes solely from the head model, which is frequency-independent [94,95,98].

Note that the lead field $\mathbf{M}(\mathbf{r}, \mathbf{r}_n)$ can be interpreted as the electric field resulting from tES [8,52,53]; the amplitude of this field along the cortical normal direction ($E_{cn}$), is related to the amplitude of the somatic $V_m$ response through a (frequency-dependent) conversion factor (Fig 6J, two different color scales).

As a final control, we directly simulate the somatic $V_m$ response of the considered neuron model (Fig 6A) under the influence of the uniform electric field given by the head model (Fig 6J) at the previously used cortical location (Fig 6E).

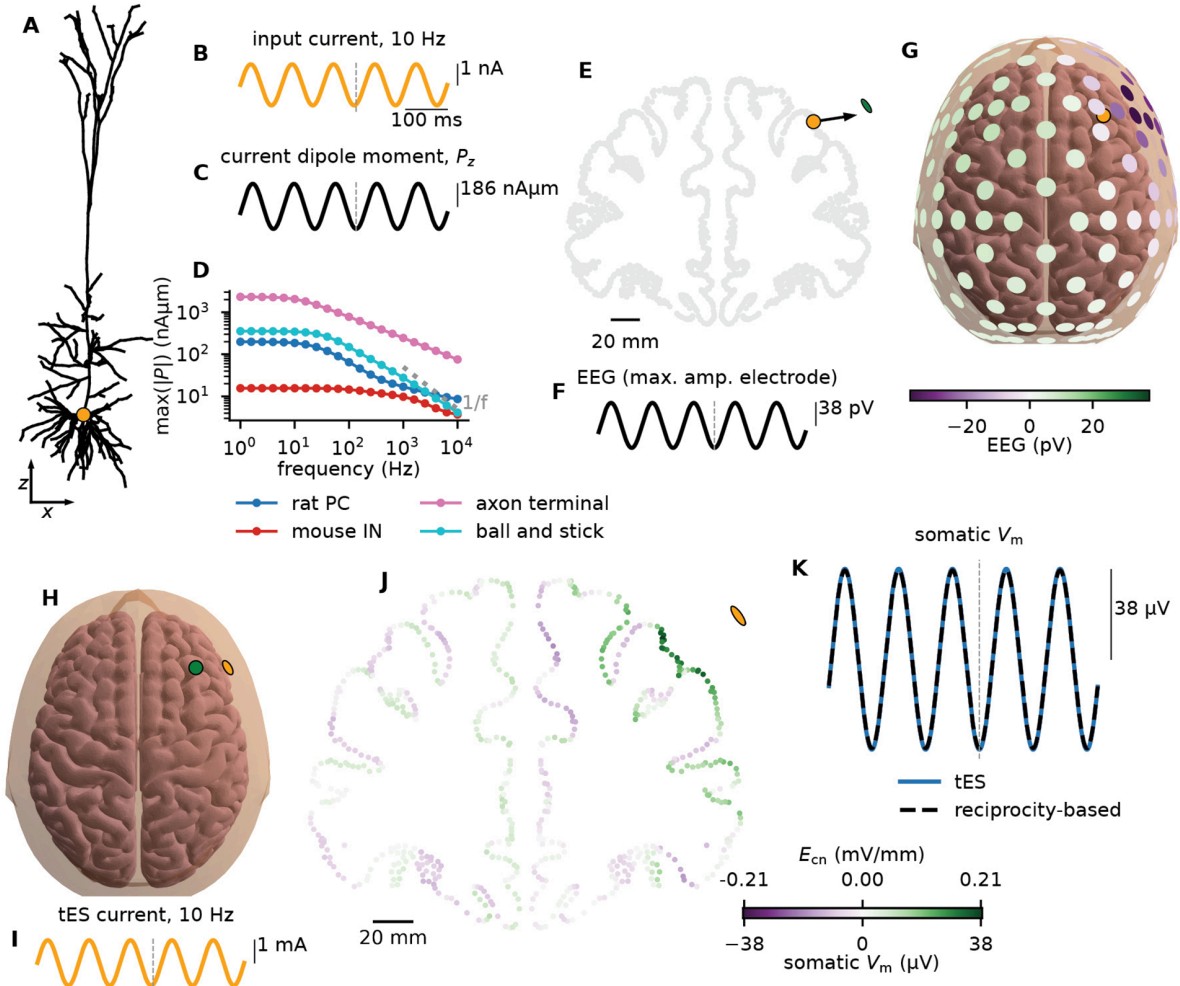

**Fig 6**. **The effect of transcranial electric stimulation (tES) on individual pyramidal cells. A:** A passive pyramidal cell model receiving a somatic current input (orange dot). **B:** The input current with a frequency of 10 Hz and a 1 nA amplitude. **C:** The $z$-component of the resulting current-dipole moment, perpendicular to the cortical surface. **D:** The amplitude of this $z$-component as a function of the input frequency for different cell models. **E:** The current-dipole (panel C) is placed in the middle frontal gyrus of the New York head model. The orange dot marks the location, the black arrow marks the orientation, and the green ellipsoid marks the nearest EEG electrode. **F:** The induced voltage signal of 38 pV at the nearest EEG electrode (green ellipsoid in panel E), from the single pyramidal cell at the location marked in panel E. **G:** The EEG signal at the time of the maximum signal (time indicated by gray dashed lines in panels B, C, and F). The location of the dipole is marked by the orange dot (partially covered by an EEG electrode). **H:** Through the RT we can invert the situation and calculate the somatic $V_m$ for the pyramidal neuron (located at the green dot) responding to tES (located at the orange ellipsoid). **I:** A 10 Hz, 1 mA current input through an EEG electrode (marked in panel E and H), that is, $10^6$ times larger amplitude than the original intracellular current input. **J:** A cross section of the head model's lead field, displaying the amplitude of the electric field along the cortical normal direction ($E_{cn}$). For a tES current amplitude of 1 mA through the considered electrode (orange ellipsoid), the maximum value of $E_{cn}$ is 0.21 mV/mm. This is the unmodified lead field as given by the New York head model [52]. The somatic $V_m$ response for the cell model in panel A at different locations in the cortex, in response to the tES in panels H and I, can be found by a scaling of the lead field, in this case by a factor of 186 µm. **K:** The somatic $V_m$ response to tES for the cell model in panel A at the location marked in panel E is $10^6 \cdot$ 38 pA = 38 µV. As a control, we also simulate the effect of a uniform electric field of amplitude 0.21 mV/mm along the $z$-axis of the cell model, and see that the somatic $V_m$ response is indistinguishable.

This approach assumes that the electric field is relatively uniform over the spatial extent of the neurons, and is sometimes called the quasi-uniform assumption [45,99]. As for the infinite homogeneous medium (Fig 2), the direct simulation of the effect of tES (Fig 6K, blue line) and the reciprocity-based approach (Fig 6K, dashed black line) yield indistinguishable results.

# 3 Discussion

We here use the reciprocity theorem (RT) to shed light on the neural response to extracellular current stimulation (ES) (Fig 1). No error is introduced using RT to estimate the effect of ES on passive cell models; errors are minimal (a few percent) for active cell models in the relevant regime, which is the subthreshold one (Fig 2). We use the RT-based approach to study how different neural elements are affected by ES (Fig 3). Furthermore, we derive an analytic expression for the somatic $V_m$ response of a passive ball-and-stick model under the influence of ES, extract simple power laws for its dependency on frequency and distance, and compare the sensitivity of ES for different neural parameters (Fig 4). We also demonstrate that the RT-based approach can be used to calculate the isolated effect of the ES itself, which can afterwards be added to study non-linear neural activity (Fig 5). Finally, we demonstrate that the RT is applicable to study transcranial electric stimulation (tES): Using a detailed head model we can directly predict the subthreshold $V_m$ response to tES of an arbitrary cell model at any location in the brain (Fig 6).

## 3.1 What is new?

The RT has a long history in neuroscience, yet its ability to directly study ES has been overlooked: It has so far only been used to swap the locations of extracellular current sources (or current dipoles), and extracellularly measured potentials, without explicitly considering the neurons themselves. We here show that through placing an electrode in the soma of a neuron model, we can use the RT to directly study ES. We demonstrate that applying the RT to study ES has many advantages: We show that the RT explains the surprising cell-type and frequency-independent ES results presented by Lee et al. (2024), and nuances these results by demonstrating that they are only expected to hold for nearby neurons below about 100 Hz. Furthermore, the RT-based approach supplies us with analytic formulas and power laws for both the frequency-response and distance-dependence of ES, which has not, to the best of our knowledge, been reported previously.

In general, moving from intracellular stimulation of a neuron to predicting the resultant $V_e$ at $n$ locations to injecting an extracellular current, as in transcranial ES, and estimating the effect on $V_m$ at $n$ location, has the same computational complexity in terms of number and order of equations to be solved. The reason for why the RT provides a fast and intuitive way of understanding ES, is that electrophysiology has a biased view of neurons, emphasizing the location and size of the cell body, at the expense of other neuronal regions: the soma is far more accessible to intracellular stimulation and recording than other neuronal compartments and is in close proximity of the axon initial segment where $V_m$ is transduced into one or more spikes. This soma-centric view has led to the development of a mature understanding regarding the origin of extracellular potentials following somatic current injections. The RT also supplies a new perspective on ES which makes it easier to understand how different neural features, like morphology and diameters, influence a neuron's susceptibility to ES. We therefore conclude that the RT-based approach is capable of providing new general insights into how ES affects neurons.

## 3.2 When to apply the RT-based approach for studying ES

The RT allows us to rely on the extensive knowledge available on how neural activity generates extracellular potentials [20], to gain a better intuition of how ES works, instead of the traditional approach which involves explicitly simulating each specific combination of stimuli and cell model, including the locations and time course of the stimuli. In the RT-based approach, a single simulation of a somatic white noise current input with simultaneous calculation of $V_e$ at all relevant extracellular locations, is sufficient to obtain the subthreshold response of that particular cell model to an ES at any one of the considered locations. Furthermore, since the system is linear and we have the full frequency response, we can, through Fourier analysis, find the subthreshold somatic $V_m$ response to an arbitrary time course of the ES, including for mono- or bi-phasic pulses of arbitrary durations.

We here considered the effect of ES on one neuron; intriguingly, our results show that the linear $V_m$ response of a set of $N_n$ neurons at different locations, resulting from a set of $N_i$ stimulation electrodes can be formulated as a matrix equation in frequency space $\mathbf{V}_m = \mathbf{M}\mathbf{I}_{stim}$, where $\mathbf{M}$ is the lead field which through the RT can be found by computing extracellular potentials due to current stimulation of the soma. We can tailor $\mathbf{I}_{stim}$ towards a desired $\mathbf{V}_m$ response in the $N_n$ neurons.

Note that the RT has been used for optimizing tES with the goal of maximizing the electric field at a given location [52–56]. However, these studies did not use intracellular currents, and were therefore, in practice, agnostic to the neurons and their properties themselves. This approach does not, for example, distinguish between tDCS and tACS since the spatial properties of the induced electric fields are typically assumed to be frequency-independent. However, tDCS and tACS can be expected to affect neurons differently. The possibility to directly optimize for a cell-type specific $V_m$ response at a given frequency via the RT is a key step towards more controllable tES.

### 3.3 Transcranial electric stimulation

Grossman et al. (2017) [100] directly induced spikes in mice with current amplitudes of ca. 0.5 mA. Yet the same current amplitudes would give two orders of magnitude lower electric field amplitudes in the human brain [30,101]. This is consistent with predictions via the RT: Halnes et al. (2024) [20] showed that EEG signals in a mouse head model were two orders of magnitude larger than EEG signals in a human head model from the same neural source. The primary reason is the 10-fold larger distance between the scalp and neural sources, translating into a 100-fold smaller $V_e$ in people and mice (based on the dipole decay of $1/r^2$; Fig 4A). Increasing currents by a 100-fold to compensate for this signal loss is problematic; indeed, such large currents were historically employed to induce electric anesthesia [11,33]. Our results are consistent with the consensus that observed effects of tES in humans must come about indirectly, either by enhancing spike synchrony in specific frequency ranges or from mechanisms like stochastic resonance [11,30,33,34].

It was previously suggested, as a rule of thumb, that an electric field of 1 mV/mm is sufficient to entrain neural activity [95,102–104]; however, neural effects of field strengths as low as 0.2 mV/mm are well documented [11,60,96,105–107]. In the detailed New York head model applied here, a 1 mA tES yields electric fields of 0.2 mV/mm (Fig 6J). tES amplitudes of 1-5 mA should therefore be in the lower range of what has been reported to affect human neural activity. We estimate that this corresponds to somatic membrane potential responses on the order of 40-200 μV at 10 Hz, consistent with previous results from both simulations and experiments [45,59,96].

That the effect of tES on membrane potentials is much smaller than the spiking threshold has implications for how tES is mediated: Axon terminals were here found to be about an order of magnitude more sensitive to ES than pyramidal cells (Figs 3 and 6, [45]); however, the membrane potential of axon terminals is expected to lie stably around rest (when not firing an action potential), and are therefore less likely to be influenced by minor induced changes in their membrane potential than somas, where the membrane potential occasionally fluctuates close to the firing threshold. Therefore, while axons are likely to be the main target of intracranial microstimulation [26,27,72], it seems more likely that the spike initiation zone is the main target of tES (but see also [46]).

### 3.4 Limitations

The principal limitation in using the RT-based approach to simulate the effect of ES is the underlying assumption of linearity, which must hold for both the neuron and the extracellular environment (brain and head). Most studies report that the electrical properties of the brain and head are approximately linear and ohmic for frequencies below several kHz [28,77,94,95,98,108,109]. Some studies report weakly frequency-dependent tissue [33,110]; for the high frequencies of temporal interference stimulation, it is plausible that such effects can become important and might challenge the quasi-static assumption [111]. Note, however, that a frequency-dependent head model would still typically be linear [18,20], and as such, the assumption of linearity with regard to the extracellular environment seems well-justified.

Neurons, on the other hand, have non-linear membrane properties, in particular when operating close to their spiking threshold. Mirzakhalili et al. (2020) [31] demonstrated how an ion-channel-mediated rectification mechanism could explain how neurons respond to the envelope frequency during TI stimulation. It is plausible that such non-linear effects are needed to explain how spikes are directly evoked by extremely high-amplitude temporal interference stimulation, such as described by Grossman et al. (2017) [100] (see also discussion in Vieira et al. (2024) [33]). If so, the RT-based approach is not suited to study this phenomenon. How weak tES-mediated $V_m$ perturbations affect neural networks, on the other hand, might be easiest to understand through simplified network simulations, where the RT-based approach can be useful (see Outlook).

Although our results here indicate that the tested neuron models behaved close to linearly in the subthreshold regime (Fig 2), some ion channels are known to be active also in the subthreshold regime. In particular, $I_h$ affects both extracellular potentials [112,113] and how neurons are affected by weak electric fields [44,91]. However, we can expect $I_h$ to primarily be important for $V_m$ in the apical dendrite. Furthermore, calcium spikes are probably more likely to be evoked when apical dendrites are depolarized [114]. Certain neuron types, for example in the basal ganglia, also exhibit intrinsic spiking behavior [75]. Therefore, in certain cases, the assumption of linearity might not be appropriate.

In this study we were primarily concerned with demonstrating how the RT can be used to understand ES, and our results are therefore more illustrative than comprehensive. We only tested a few cell models from the literature. Further studies should scan through entire cell databases from ModelDB [115], The Allen Brain Institute, or BBP [27,45,116,117], to demonstrate the applicability of the same principle to neurons of different cell types and brains of different species, which will be important for better understanding ES.

We found good agreement between our simulations and experimental data from microstimulation (Sec 2.1) and tES (Sec 2.6). The main validation of the RT-based approach is, however, the demonstration that it provides results equivalent to the traditional approach to simulating brain stimulation in the subthreshold regime. The validity of the reciprocity-based approach is therefore anchored to the traditional approach, which has been rigorously validated (see Aberra et al. (2025) [46] for a recent example).

## 3.5 Outlook

Ours and previous results indicate that the direct effect of tES on the membrane potential of individual neurons is weak, in the range of ten to a few hundreds of μV, far away from the threshold for spiking. This implies that any behavioral and cognitive effects of tES must be mediated by synergistic network effects, such as stochastic resonance and/or by advancing or delaying spike times [11,30,34,41,96].

Our study focuses on the single-cell level; however, to understand the effect of ES on the brain, it is vital to study network effects. In general, it is not feasible to predict these by extrapolating single-cell responses. However, accurate simulations of the network effects of ES must be built on an accurate representation of single cell effects. To study the effect of ES at the network level, one can, in principle, use large-scale network models of biophysically detailed multicompartment cell models, and simulate the ES with traditional methods. However, due to their associated high computational demands, such models are challenging to work with. Furthermore, comprehensive parameter scans are not within reach.

A common way to study network effects is via point-neuron networks or mean-field models [118], including the effects of ES/tES [104,107,119–122]. However, the effects of ES are mediated by electric potential gradients over neural morphologies, which are not part of such approaches. It is here that the RT-based approach can be used to estimate the somatic membrane potential response to ES with biophysically detailed cell models, which can then be added to the membrane potential of simplified neuron models [123] or point neurons in a network model.

The assumption of linearity is therefore not as restrictive as it might first seem: The RT-based approach is highly accurate in estimating the effect of ES in the subthreshold regime (Figs 2E, 5F, and 6K), and the point-neuron models account for the non-linear spiking behavior. The RT-based approach incorporates the detailed cellular morphology, and can

therefore be expected to be more accurate than approaches using simplified or reduced morphologies for capturing the effect of ES. Furthermore, it will be more computationally efficient, since the network activity can be simulated with point-neurons rather than neuron models having a spatial structure. The RT-based approach can directly predict the somatic $V_m$ effect of ES throughout a volume, for different cell types, and under different stimulation parameters, which can be used with point-neuron network simulations using NEST [124], or even whole-brain mean-field models using The Virtual Brain [125].

## 4 Methods

### 4.1 Neural simulations

All simulation code was written in Python, and neural simulations were done through LFPy 2.3 [79] running on NEURON 8.1 [126].

### 4.2 Cell models

The pyramidal cell model that was used was the rat cortical layer 5 cell model presented by Hay et al. (2011) [63]. The interneuron model was a perisomatic layer 5 aspiny PV cell from the mouse primary visual area from the Allen Brain Atlas (celltypes.brain-map.org/; electrophysiology ID: 469610831; model ID: 491623973). The parameters for the myelinated axon model were from [127], in the implementation used by [128]. The total length was 10 mm, and a node of Ranvier was inserted for every 50 μm. We also confirmed that results were similar for the myelinated axon model created by [42]. The axon terminal was just an axon endpoint with the same diameter as the rest of the axon.

Passive versions of the cell models were modified versions of the original cell models, where all active conductances were removed.

### 4.3 Input to cell models

Intracellular input currents were modeled as synapse currents ("POINT_ PROCESS" in NEURON), that is, they are treated as membrane currents. This is because the membrane potential is measured across the membrane ("port 1" in Fig 7A), and when applying the RT, the current should also be across the membrane ("port 1" in Fig 7B).

For intracellular current input to active cell models (Fig 2), we relied on a hand-picked, fixed cell-specific current amplitude in the subthreshold regime. This was done because the different cell models have vastly different input impedances and react differently to the same current input. The amplitude was 0.1 nA, 0.01 nA, and 0.001 nA for the pyramidal cell model, interneuron model, and axon model, respectively. The resulting extracellular potential was afterwards scaled to reflect the amplitudes of the extracellular current source.

White-noise current input was constructed as a sum of sinusoids, where each frequency component had the same amplitude (1 nA) and a random phase [67,109,112]. The frequency components used when constructing the white noise were the same as those considered when extracting amplitude- and phase spectrums through Fourier analysis.

### 4.4 Simulating extracellular stimulation

The effect of an external extracellular potential on a neuron can be simulated with LFPy [79], running on top of the NEURON simulation environment [126].

**4.4.1 Infinite homogeneous mediums** The extracellular potential at the midpoint location of each neural compartment is calculated for each time step, and inserted into NEURON as a boundary condition through the extracellular mechanism. For infinite homogeneous mediums, the extracellular potential from an ES point source $I_{stim}(t)$ at location $\mathbf{r}'$ can be calculated as [20],

$$V_{EC}(\mathbf{r}, t) = \frac{I_{stim}(t)}{4\pi\sigma_t|\mathbf{r} - \mathbf{r}'|} \ . \tag{5}$$

**4.4.2 Detailed head model** Since the tES electrodes are far away relative to the size of neurons, the induced electric fields will be relatively uniform over the extent of individual neurons. We therefore used the quasi-uniform assumption [45,99], calculating the resulting extracellular potential along the z-axis, as $V_{EC} = E_{cn}z$. Here, $E_{cn}$ is the electric field along the cortical normal direction (the z-axis of the local coordinate system), which we obtain directly from the lead field matrix of the New York head model [8,52].

The New York head model takes into account the detailed geometry of the head, in addition to the very different electric conductivities of the white matter, gray matter, air cavities, cerebrospinal fluid, skull, and scalp, with conductivity values of 0.276 S/m for gray matter, 0.126 S/m for white matter, 1.65 S/m for CSF, 0.01 S/m for the skull, and 0.465 S/m for the scalp [52,129]. This model is "pre-solved" through numerical finite element simulations (FEM), where the lead field matrix **M** has been calculated and made freely available (https://www.parralab.org/nyhead/; [52]). It was constructed through current input to each one of 231 electrodes, where the electric field was calculated and stored at about 75,000 locations in the brain. The original size of the lead field matrix is 231×75,000×3. It is often convenient to simplify this by assuming that dipoles are always oriented along the local cortex normal direction, reducing the size to 231×75,000. These matrix elements can thus be interpreted either as the electric field along the cortical normal direction at any one of the 75,000 cortical locations in response to tES from any one of the EEG electrodes, or equivalently, as a map from a unit dipole at any one of the 75,000 locations (assumed to be oriented along the cortical normal direction) to the resulting EEG signal at any one of the 231 electrodes.

## 4.5 Signal analysis

For the Fourier analysis we used the `scipy.fftpack` Python package to extract the amplitude $V_e(f)$ and phase $\theta(f)$ spectrums from calculated signals. When plotting the spatial profiles of extracellular potentials at specific frequencies (Fig 3), the phases of $V_e$ were incorporated through plotting $\mathrm{Re}\{V_e(f)e^{i\theta(f)}\}$.

## 4.6 Analytic expressions

For the somatic membrane current, the transfer function is given by [73]

$$\mathbf{T}_l^s = -\frac{\sinh(\mathbf{q}l/\lambda)}{\mathbf{Y}\cosh(\mathbf{q}l/\lambda) + \sinh(\mathbf{q}l/\lambda)} \ , \tag{6}$$

and for the current-dipole moment the transfer function is given by,

$$\mathbf{T}_p^s = \frac{\lambda}{\mathbf{q}}\frac{\cosh(\mathbf{q}l/\lambda) - 1}{\mathbf{Y}\cosh(\mathbf{q}l/\lambda) + \sinh(\mathbf{q}l/\lambda)} \ . \tag{7}$$

Here $\mathbf{q}^2 = 1 + j\omega\tau_m$, $\mathbf{Y} = \frac{\mathbf{q}d_s^2}{d\lambda}$, and $l$ and $d$ is the length and diameter of the stick, respectively, while the diameter of the soma is $d_s$.

## 4.7 Application of the reciprocity theorem

One can only expect the RT to be applicable for approximately linear systems, which includes passive cell models, that is, cell models without any active conductances, and active cell models which are operating in the subthreshold regime, where neurons often behave approximately linearly [59,61,112,113,130,131]. However, the amplitude of extracellular current sources used to stimulate neurons (1000 nA in Fig 2B) is a typical amplitude that would drive neurons to fire action potentials if given intracellularly, and the neurons would therefore be far outside the approximately linear regime. Directly using the same current amplitudes for extracellular and intracellular current sources is therefore applicable for passive cell models, but not for active cell models. For intracellular current input to active cell models (Fig 2) we therefore relied on the

assumed approximate linearity in the subthreshold regime and used a fixed, cell-specific, subthreshold current amplitude (0.1 nA for the pyramidal cell model in Fig 2A). The resulting extracellular potential was afterwards scaled by the ratio of the extracellular to intracellular current amplitude (that is, by a factor of 1000 nA / 0.1 nA = 10,000 in Fig 2B). The current amplitudes used in different types of ES vary over many orders of magnitude from microampere or even less in microstimulation, to milliampere in tES. The scaling factor varies by the same amount. Note, however, that this is unproblematic as long as the membrane potential response of the neuron to ES is in the linear, i.e., subthreshold regime, in practice below a few millivolts (Fig 2). This is what matters, not the amplitude of the tES.

In electromagnetism, the reciprocity theorem states that for time-invariant linear media the relationship between an oscillating current and the resulting electric field is unchanged if one interchanges the points where the current is placed and where the field is measured [132]. When simulating neural activity, it is standard practice to represent neurons as equivalent electric circuits [21,58,133]. For the special case of electric circuits, the reciprocity theorem states that in any passive linear network, if a current source $I_0$ between nodes $x$ and $x'$ produces the voltage response $V_y$ between nodes $y$

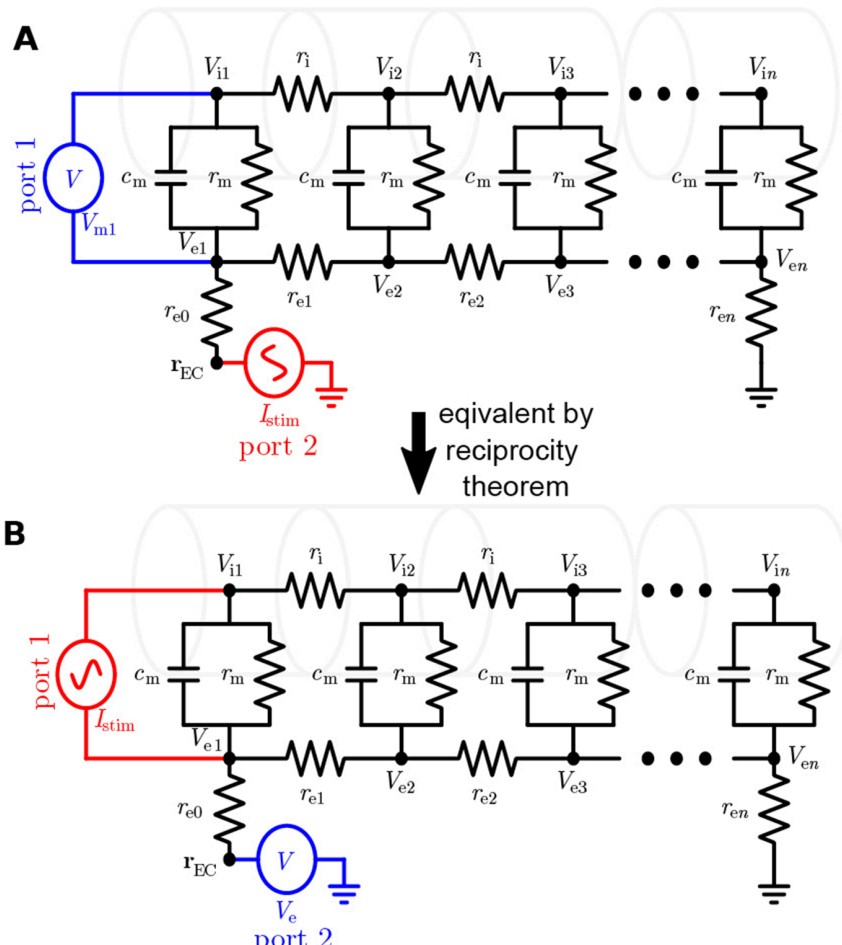

**Fig 7. Illustration of the reciprocity theorem. A:** The equivalent circuit diagram for a passive neuron model stimulated by an extracellular current source $I_{stim}$, while the membrane potential of compartment 1 is measured. **B:** The equivalent circuit diagram for a passive neuron model stimulated by an intracellular current source $I_{stim}$, while the extracellular potential is measured. According to the reciprocity theorem, the measured potential in the two equivalent circuits should be the same.

and $y'$, then the removal of the current source from nodes $x$ and $x'$ and its insertion between nodes $y$ and $y'$ will produce the voltage response $V_y$ between nodes $x$ and $x'$ [132].

In the context of electric stimulation of neurons, this implies that if there is a current source $I_{stim}$ at extracellular location $\mathbf{r}_{EC}$, flowing to ground (Fig 7A "port 2"), and this produces the voltage response $V_{rec}$ over a given part of a neural membrane (Fig 7A "port 1"), then moving the current source to instead be across the neural membrane (Fig 7B "port 1") will produce the same voltage response $V_{rec}$ between location $\mathbf{r}_{EC}$ and ground [132] (Fig 7B "port 2").

The reciprocity theorem is valid for any electrical network that consists entirely of ideal capacitances, inductances, and resistances, that is, passive electrical networks. For simplicity, one can start by analyzing the electrical network in the absence of the resting membrane potential, because the principle of superposition ensures that the resting membrane potential can be added afterwards as additional voltage sources ("batteries") in the equivalent circuit. When the resting membrane potential is not taken into account, the RT-based approach only predicts deviations of $V_m$ from the resting membrane potential.

Because of the conventions regarding the direction of membrane currents, a sign change is needed for the intracellular current input (relative to the extracellular current source) when it is inserted in NEURON as a synapse current.

## Author contributions

**Conceptualization:** Torbjørn Vefferstad Ness, Christof Koch, Gaute T. Einevoll.

**Data curation:** Torbjørn Vefferstad Ness.

**Formal analysis:** Torbjørn Vefferstad Ness, Gaute T. Einevoll.

**Funding acquisition:** Gaute T. Einevoll.

**Investigation:** Torbjørn Vefferstad Ness.

**Methodology:** Torbjørn Vefferstad Ness.

**Project administration:** Torbjørn Vefferstad Ness, Christof Koch, Gaute T. Einevoll.

**Resources:** Torbjørn Vefferstad Ness, Gaute T. Einevoll.

**Software:** Torbjørn Vefferstad Ness.

**Supervision:** Christof Koch, Gaute T. Einevoll.

**Validation:** Torbjørn Vefferstad Ness.

**Visualization:** Torbjørn Vefferstad Ness.

**Writing – original draft:** Torbjørn Vefferstad Ness, Christof Koch, Gaute T. Einevoll.

**Writing – review & editing:** Torbjørn Vefferstad Ness, Christof Koch, Gaute T. Einevoll.

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
