## [Decision Letter · Decision Letter 0]

29 Jul 2025

PCOMPBIOL-D-25-01091

Predicting neural responses to intra- and extra-cranial electric brain stimulation by means of the reciprocity theorem

PLOS Computational Biology

Dear Dr. Ness,

Thank you for submitting your manuscript to PLOS Computational Biology. After careful consideration, we feel that it has merit but does not fully meet PLOS Computational Biology's publication criteria as it currently stands. Therefore, we invite you to submit a revised version of the manuscript that addresses the points raised during the review process.

Please submit your revised manuscript within 60 days Sep 28 2025 11:59PM. If you will need more time than this to complete your revisions, please reply to this message or contact the journal office at ploscompbiol@plos.org. Please include the following items when submitting your revised manuscript:

We look forward to receiving your revised manuscript.

Kind regards,

Christian Keitel

Academic Editor

PLOS Computational Biology

Daniele Marinazzo

Section Editor

PLOS Computational Biology

**Additional Editor Comments:**

Dear Authors,

We've had the pleasure of having three experts in the field submit their comments on your manuscript. You will find that all of them express their support for, and see merit in the work. Two of them express strong ideas on how you can further increase the impact of your findings, by adding stronger validations and/or by bridging neuronal and network levels more strongly.

We are looking forward to your revisions,

Christian Keitel

**Journal Requirements:**

At this stage, the following Authors/Authors require contributions: Torbjørn Vefferstad Ness, Christof Koch, and Gaute T. Einevoll. Please ensure that the full contributions of each author are acknowledged in the "Add/Edit/Remove Authors" section of our submission form.

Potential Copyright Issues:

i) Figure 5. Please confirm whether you drew the images / clip-art within the figure panels by hand. If you did not draw the images, please provide (a) a link to the source of the images or icons and their license / terms of use; or (b) written permission from the copyright holder to publish the images or icons under our CC BY 4.0 license. Alternatively, you may replace the images with open source alternatives. See these open source resources you may use to replace images / clip-art:

**Reviewers' comments:**

Reviewer's Responses to Questions

**Comments to the Authors:**

Reviewer #1: This manuscript presents an interesting and potentially valuable application of the reciprocity theorem to the field of electrical brain stimulation. The core idea – leveraging the RT to simplify and better understand the relationship between stimulation parameters and neuronal responses – is clear. The manuscript is generally well-written, and the authors demonstrate a good grasp of the relevant biophysical principles. I believe the work has the potential to be publishable in PLOS Computational Biology, but requires Major Revision to address several key concerns regarding rigor, clarity, and impact. The current draft leans towards being a methodological description with promising implications, rather than a conclusive study with significant new biological insights. Please find the point-by-point feedback below.

Relation to the Biological Problem: The authors utilized a largely passive (non-spiking) single-cell model, representing the simplest possible case and one well-explored in the literature. The next steps for such modeling remain unclear. Traditionally, this would involve embedding a single cell into a spiking network representing a cortical column or brain region. However, the feasibility of this approach is uncertain, as network dynamics can quickly become chaotic and non-linear. The authors should consider whether their approach has a future in addressing more realistic, network-level modeling.

Limited Biological Validation & Scope: The manuscript heavily relies on simulations. While simulations are essential, the lack of direct comparison to a wider range of experimental data, beyond the reference to Lee et al. (2024), significantly weakens the impact. The current analysis appears largely hypothetical. Where are the predictions that differentiate this approach from existing modeling strategies and can be experimentally tested? A strong addition would be a validation against a broader set of established experimental paradigms, or a demonstration of a novel prediction that can be directly tested.

Validation vs. Verification: On a related note, the authors claim to have “validated” the approach by comparing simulations against other simulations, which is an overreach. True validation requires generating testable predictions and experimentally validating them. What the authors have performed is more accurately described as a sanity check or a first approach verification.

Overstatement of Novelty: The authors present the application of the RT to this specific context as novel. While the application of electrical stimulation is relatively unexplored, the RT itself is well-established and used in both EEG and brain stimulation (e.g., for stimulation electrode location optimization). Furthermore, numerous preceding works exist that generated similar insights into single-cell behavior due to electric current/field injection. The authors present no clear reason to prefer their implementation over preceding approaches. The introduction needs to better acknowledge the existing use of the RT in related neurophysiological modeling. The authors should be more specific about what is novel here – a particular application of the RT to single-cell modeling.

Clarity of Practical Implications for tES: While the manuscript highlights the potential to predict somatic membrane potential changes due to tES, the connection to real-world scenarios is weak. For example, the authors injected the electric current at the soma of their model – in reality, the electric field is much larger than the neuron and affects all compartments simultaneously.

Insufficient Justification for Specific Parameters/Models: The choice of specific cell models (rodent's L5 pyramidal and interneuron, an axon) feels somewhat arbitrary. What is the rationale behind this selection, and how generalizable do the authors expect the results to be to different neurons of different species (particularly, humans)? A more systematic exploration of the impact of different cell morphologies and biophysical properties on the results would strengthen the conclusions.

Reviewer #2: In “Predicting neural responses to intra- and extra-cranial electric brain stimulation by means of the reciprocity theorem” (PCOMPBIOL-D-25-01091), Ness et al. use the "Reciprocity Theorem" to predict the effects of electrical brain stimulation, and in particular, transcranial electrical stimulation.

The reciprocity theorem is a physical principle that allows one to exchange the location of current and voltage measurements within (most) circuits. Although this theorem is fairly straightforward—and used extensively in other fields, it has seen limited use in neuroscience and this is the first time I have seen it used to explore brain stimulation. Except for a few points, below, the paper generally seems technically sound. It is extremely clearly written with lovely illustrations.

Its major weakness is that I’m not sure what new insight this method adds for practitioners of brain stimulation. The major conclusion (§3.3, 3.5) is that the depolarizing effects of transcranial electrical stimulation are far too weak to directly drive spiking. This is already the overwhelming consensus from computational and experimental studies; indeed, the field has struggled too how that The real mystery, as the authors note on p. 15, is how network mechanisms translate these small polarizations into surprisingly robust changes in (e.g.,) spike timing seen in animal models and BOLD. The paper briefly touches on this at the top of p. 14-15. If the authors could expand on this and offer candidate mechanisms, it would be a really impactful paper. Otherwise, it’s essentially a editorial judgement: this uses a standard (but under-appreciated) to re-confirm widely accepted results, but it is an interesting perspective and does tie together aspects of the field that may not be obviously related.

Minor:

- The authors do an excellent job discussing several papers that (in my opinion), are both influential and often misinterpreted. The discussion of Grossman et al. (2017) should probably also reference Rampersad et al. (2019; Neuroimage), who re-modelled the mouse and found that the electric field was very, very strong (300+ V/m); note that Grossman also removed the skin, which removes a lot of the shunting. I think that, rather than the current, ought to be the focus. I don’t have much to add re: the 1 V/m but thank you for including that nuance — those early experiments conflate the (true) presence of an effect with its detectability. I would note that other estimates of the E-field in human brains suggest that up to 0.8 V/m (at 2 mA) is achievable in humans (Huang et al, 2017; ELife—but note the correction).

Temporal interference needs to be discussed carefully since there is not yet a consensus on how it acts. In the Mirzakhalili model, the demodulation is only partially effective; this is consistent with data on amplitude-modulated tACS. Are the authors sure this isn’t important? Can it somehow be addressed via RT or does it provide more of an upper bound?

On p. 3, it notes that the RT is applicable in the subthreshold regime and this approach apparently requires an ad hoc scaling factor (§4.7) to keep the model there. It feels a little circular to me to make this adjustment and then conclude that the effects, assumed to be small, are in fact small. Is there a principled way to break this loop or a diagnostic that tells you when you’re in trouble? In other words, suppose I want to simulate a 15 mA current pulse? Is it really supra threshold or have I mischosen the scale factor?

Both Aberra et al. and Vieira et al. have recent preprints examining the effects of electric fields on axons, which may be of interest to the authors.

Section 2.4 argues that implants may boost electric fields. The authors might consider discussing Mercadal et al. (2022; J Neural Engineering), who argue that metal is less of a problem than believed because of the mismatch between ionic and electronic current flow? More generally, this section is interesting but needs to be tied more tightly to the paper’s message. Is the argument that ECOG recordings overstate ||E||?

Reviewer #3: Review

In their work Ness et al. present applications of the reciprocity theorem for invasive and non-invasive electric brain stimulation to predict neural responses to stimulation. It leverages the principles of RT to infer how extracellular currents (induced by stimulation) affect intracellular potentials affecting neural behavior.

The manuscript is very well written, and provides interesting insights to the cellular effects of electric stimulation. I generally support publication of the manuscript, I only have a few remarks on the presentation of the work.

1. I am missing a description of the putative mechanisms of ES/tES (tDCS/tACS) in the introduction. All the authors mention so far is that these mechanisms of tES are controversial. However, especially at the cellular level there are many studies demonstrating effects electric stimulation (Bindman et al., 1964; Fröhlich & McCormick, 2010; Krause et al., 2019; Ozen et al., 2010; Reato et al., 2010).

2. With respect to the implications of the work for tES the manuscript has a strong focus on temporal interference stimulation (TI), which is a very novel technique. I think it is very fair to discuss it here, but I think the authors should also go into a bit more depth in terms of the implications of their work for more established tES techniques (i.e. tDCS and tACS).

3. There are some constraints in using computational headmodels like the NY head in estimating electric fields in the brain that should be mentioned. These models have to make assumptions about tissue conductivities and limit the number of compartments the head is segmented into which will affect the resulting electric field strength and distribution. This introduces uncertainty in the estimations of induced electric fields the authors discuss. I would also recommend adding a brief description of the NY head (number of compartments, assumed tissue conductivities).

4. Maybe I missed this in the manuscript, but since the RT is only applicable in the subthreshold regime, how far is it applicable in the context of an active neuron model with intrinsic spiking behavior? Here the intrinsic membrane potentials should frequently reach a non-linear regime.

**Have the authors made all data and (if applicable) computational code underlying the findings in their manuscript fully available?**

Reviewer #1: Yes

Reviewer #2: Yes

Reviewer #3: Yes

PLOS authors have the option to publish the peer review history of their article (what does this mean?). If published, this will include your full peer review and any attached files.

Reviewer #1: No

Reviewer #2: No

Reviewer #3: No

**Figure resubmission:**
---

## [Decision Letter · Decision Letter 1]

19 Oct 2025

PCOMPBIOL-D-25-01091R1

Predicting neural responses to intra- and extra-cranial electric brain stimulation by means of the reciprocity theorem

PLOS Computational Biology

Dear Dr. Ness,

Thank you for submitting your manuscript to PLOS Computational Biology. After careful consideration, we feel that it has merit but does not fully meet PLOS Computational Biology's publication criteria as it currently stands. Therefore, we invite you to submit a revised version of the manuscript that addresses the points raised during the review process.

Please submit your revised manuscript within 30 days Dec 19 2025 11:59PM. If you will need more time than this to complete your revisions, please reply to this message or contact the journal office at ploscompbiol@plos.org. Please include the following items when submitting your revised manuscript:

We look forward to receiving your revised manuscript.

Kind regards,

Christian Keitel

Academic Editor

PLOS Computational Biology

Daniele Marinazzo

Section Editor

PLOS Computational Biology

**Additional Editor Comments :**

Dear Authors, one of the reviewers has added more sensible suggestions based on your revision that we would like you to consider.

**Reviewers' comments:**

Reviewer's Responses to Questions

Reviewer #1: The authors adequately addressed my points. I recommend this article for publication and am looking forward to more work from the authors demonstrating the practical use of RT in clinical neuroscience.

Reviewer #2: Thank you for responding to my comments on the previous version. I do have a further few suggestions that I'd like the authors to consider.

1a) I still find the "novelty" aspect overstated and the new Figure 1 doesn't really help much, at least in my opinion. That said, I think there's a lot of value in the concept of reciprocity as a way to unite other findings in the literature. I would consider removing that Figure and slightly editing the introduction.

1b) I'd appreciate a more consolidated summary of the Lee et al. data: . i.e. something like "Lee et al. found X, which was unexpected b/c Y. Our model predicts X'..." I understood Figure 3 to be model vs. model predictions and the validation against empirical data sorta got lost in the shuffle.

3) I didn't mean to force the authors to totally disclaim applications to TI; I just thought it would be important to note that the mechanism may be different and thus over-estimate the effects.

4) The revised text helps a lot. Perhaps the root of my confusion was over what "sub-threshold" means. As I understand it, neurons embedded in an intact, awake brain generally sit within a few mV of their firing thresholds and so small changes in Vm can elicit "extra" spikes even if they aren't bringing a neuron all the way from it's equilibrium potential to firing. Would it be possible to include some rough numbers: "remains within the linear regime (~X mV from spike threshold)"?

6) Thanks. This is one of the examples that made the RT "click" for me.

I would add an additional caution here, since the authors mention predicting cell-type effects. While these are well-predicted by conventional models and in vitro experiments (e.g., Radman et al., 2009), I do not believe anyone has successfully identified them in vivo (Ozen et al., 2010; Krause et al., 2019, 2022; Johnson et al., 2020, etc). I don't think any of these are studies are necessarily wrong. Instead, network effects may swamp the effects of morphology: interneurons are less polarizable but more interconnected. Thus, the RT (or other isolated-neuron models) can tell you how stimulation directly affects a neuron but may not accurately describe the net effect on the brain. This point needs to be made very clearly!

Reviewer #3: The authors addressed my comments very well. I support publication of the manuscript.

**Have the authors made all data and (if applicable) computational code underlying the findings in their manuscript fully available?**

Reviewer #1: Yes

Reviewer #2: Yes

Reviewer #3: Yes

PLOS authors have the option to publish the peer review history of their article (what does this mean?). If published, this will include your full peer review and any attached files.

Reviewer #1: No

Reviewer #2: No

Reviewer #3: No

**Figure resubmission:**
---

## [Decision Letter · Decision Letter 2]

24 Nov 2025

Dear Dr. Ness,

We are pleased to inform you that your manuscript 'Predicting neural responses to intra- and extra-cranial electric brain stimulation by means of the reciprocity theorem' has been provisionally accepted for publication in PLOS Computational Biology.

Best regards,

Christian Keitel

Academic Editor

PLOS Computational Biology

Daniele Marinazzo

Section Editor

PLOS Computational Biology

Reviewer's Responses to Questions

**Comments to the Authors:**

Reviewer #2: Thank you for considering my suggestions over the last two rounds. I believe the paper is suitable for publication in PLOS Biology.

**Have the authors made all data and (if applicable) computational code underlying the findings in their manuscript fully available?**

Reviewer #2: Yes

PLOS authors have the option to publish the peer review history of their article (what does this mean?). If published, this will include your full peer review and any attached files.

Reviewer #2: No

---

## [Editor Report · Acceptance letter]

PCOMPBIOL-D-25-01091R2

Predicting neural responses to intra- and extra-cranial electric brain stimulation by means of the reciprocity theorem

Dear Dr Ness,

I am pleased to inform you that your manuscript has been formally accepted for publication in PLOS Computational Biology. Your manuscript is now with our production department and you will be notified of the publication date in due course.

With kind regards,

Narmatha Raju, M.Sc
